# Economic Improvement of Artisanal Fishing by Studying the Survival of Discarded *Plectorhinchus mediterraneus*

**DOI:** 10.3390/ani12233423

**Published:** 2022-12-05

**Authors:** Ignacio Ruiz-Jarabo, Blanca Partida, María Page, Diego Madera, Nuria Saiz, Aitana Alonso-Gómez, Lisbeth Herrera-Castillo, Esther Isorna, Ángel L. Alonso-Gómez, Ana I. Valenciano, Nuria de Pedro, Jorge Saez, Maria J. Delgado

**Affiliations:** 1Departament of Genetics, Physiology and Microbiology, Faculty of Biological Sciences, University Complutense, 28040 Madrid, Spain; 2SolDeCocos (Society for the Development of Coastal Communities), 41003 Seville, Spain

**Keywords:** discards, physiology, rubberlip grunt, small-scale fisheries, stress, survival

## Abstract

**Simple Summary:**

Artisanal fishing constitutes a subsistence activity in much of the world. Changes in fishing legislation can damage the sector if rapid adaptations are not achieved. As Europe demands landing of all catches, except those whose survival is scientifically proven, this study proposes possible improvements for the sector through a pilot study. We evaluated the survival of discarded fish and the possible recapture once they have reached a commercially profitable size.

**Abstract:**

Europe calls for the end to fisheries discards, which means bringing all caught fish (subject to minimum sizes or quotas) to land. This decision is beneficial to the ecosystem, since it forces the selectivity of the fishing gears to improve. However, artisanal fishermen find themselves in a vulnerable situation where their subsistence depends on catches with small profit margins. An exemption to this landing obligation exists, as it is also ruled that those animals whose survival is scientifically guaranteed may be returned to the sea. Here we study the survival of *Plectorhinchus mediterraneus* captured by hookline and gillnet, as well as their physiological recovery. Survival exceeds 93% in both cases. The physiological assessment of primary (cortisol) and secondary (energy mobilization, acid-base and hydromineral balance, and immune system) stress responses indicates that surviving animals are able to recover after fishing. Thus, we propose the optimal size of capture of this species to achieve greater economic benefit. For this, we rely on the prices according to size in recent years, as well as on the growth curves of the species. In this way, by releasing fish of less than 1 kg, the current benefits could be multiplied between 2.3 and 9.6 times. This pilot study lays the groundwork for regulating artisanal fisheries through scientific data related to survival of discards along with information on the sale prices.

## 1. Introduction

Small-scale fisheries (SSF) are roughly defined as those using small, underpowered boats and fishing gears that can be changed to suit different species throughout the year [1,2,3,4]. Artisanal fisheries are SSF where simple or traditional fishing methods are used, and focus on both subsistence and commercial fishing [3,5]. These fisheries are being displaced by large-scale fisheries (mainly trawling, purse seining and long-lining), which consume more natural resources and threaten the future of extractive fishing in the ocean [6]. Artisanal fisheries perform traditional activities which have been active for hundreds or even thousands of years, managed by stakeholders and the fishermen themselves, and seek (to a certain degree) the sustainability of fishing [3,7]. Compliance with the three basic pillars of sustainability requires a balance among social, economic and ecological development [8], and that is why current policies aim to promote artisanal fishing [9].

In Europe, SSF are defined as “fishing carried out by vessels up to 12 m in length and not using towed gear” [10]. Thus defined, by 2019, the SSF fleet in Europe was made up of 42.838 vessels, 58% of the total number of active fishing vessels [11]. It provides direct employment to about 62,650 people, which makes it a highly socio-economic relevant activity [12]. In southwestern Europe, SSF account for around 92% of the total fishing fleet in Portugal [13], 80% in France [14], 70% in Italy [15], and 73% in Spain [16]. The fragility of this sector makes it profoundly affected by economic and social changes, such as those related to the COVID-19 pandemic [17], or the rise in fuel prices/the removal of fuel tax exemptions [18,19].

In 2019, the European Regulation EU N° 1380/2013 became effective, establishing that all captured species subjected to catch limits or minimum sizes should be landed. This measure can affect the economy of artisanal fleets, limiting their catches and forcing them to make changes in their fishing gears to improve their selectivity. However, as stated in Article 13 of the regulation, captured animals are allowed to be released back into the sea if robust scientific evidence indicates high survival rates. This evidence must be collected for each species, fishing gear and geographical area. Since then, there have been numerous studies focused on evaluating the survival of catches, including teleosts [20,21,22], elasmobranchs [23,24], cephalopods [25] and crustaceans [26,27]. Based on those studies, the European Commission approved the first exemption to this landing obligation (EU Commission Delegated Regulation 6794/2018) for *Pagellus bogaraveo* caught using the voracera fishing gear in the Strait of Gibraltar (SW Europe). It is worth mentioning that the progress made by these latest works consisted in evaluating not only the survival, but also the physiological recovery of the animals after fishing.

It has been reported that capture by fishing is a challenging stressor to aquatic organisms [20,23,28,29,30], and that organisms must react through a series of physiological responses to overcome it. In vertebrates, primary stress responses include the release of catecholamines [31] and corticosteroids [28,32] to the bloodstream, cortisol being the most important corticosteroid in teleost fish [33]. Secondary stress responses involve physiological reactions promoted by these hormones, and include changes in the cardio-respiratory system, mobilization of energy metabolites into the blood, changes in the acid-base and hydromineral balance, and in the immune system [20,25,26,28,34]. Glucocorticoids induce the release of glucose (amongst other metabolites, such as proteins and amino acids) after capture, leading to high circulating concentrations of lactate due to an increased anaerobic metabolism [21,28,32,35]. The high levels of plasma CO_2_ and lactate induced by an acute stressor promoted blood acidification [28,34], counteracted by peripheral tissues. Due to the vasodilation of blood vessels, osmoregulatory disturbances also occur after fishing [20,28,34,36], and endocrine responses are activated to restore ionic homeostasis [28,37,38,39]. According to the existing literature, marine animals manage to recover from an acute stress situation caused by fishing in less than 24 h [20,23,25,26,32,40,41]. If the stressful situation continues over time or if a multiple system failure occurs, tertiary stress responses are triggered, leading to weakened immune responses, reduced growth, reproductive and behavioral impairments, and even death [42,43].

Rubberlip grunt (*Plectorhinchus mediterraneus*) is a teleost fish species (Family Haemulidae) distributed in the Eastern Atlantic (from Spain to Namibia) and the Western Mediterranean Sea. It can grow up to 80 cm in length and 7.5 kg in weight (www.fishbase.org). In southwestern Europe (Spain and Portugal) and the Eastern Central Atlantic area, it is an economically relevant species at the local level, especially in artisanal subsistence fishing [44]. It is highly appreciated in the Gulf of Cadiz (south of Spain), where it can reach prices of EUR 10 per kg at first sale.

The aim of this study was to develop a methodology that allows the economy of the artisanal fishing sector to improve based on the survival of discards. For this reason, we focused on the Conil fleet (a town in the south of Spain devoted to artisanal fisheries), evaluating the survival and physiological recovery of rubberlip grunt captured by hookline and gillnet, and calculating the price that released and recaptured animals could reach according to the growth rates of this species.

## 2. Materials and Methods

### 2.1. Ethics Statement

This study did not involve endangered or protected species, and it used samples from non-living animals obtained by fishermen during their usual commercial practices. Thus, it was performed in accordance with the Guidelines of the European Union (2010/63/UE) and the Spanish legislation (RD 53/2013) for the use of laboratory animals, and was approved by the Ethics Committee of the University Complutense CEA-UCM (CEA 09/03/2022).

### 2.2. Geographical Location of the Fisheries, Vessels, and Fishing Gears Characteristics

Fish were collected from hookline and gillnet in the fishing grounds of the artisanal fleet of Conil, in the Gulf of Cadiz (South of Spain, Figure 1). The position of each set was recorded using the global position system (GPS) by the fishermen and given to investigators at the end of each day. Captures were performed from November 2021 to August 2022 at depths ranging from 10 to 50 m. This study was conducted onboard the following commercial artisanal fishing vessels: *Rosario* (plate ID# 3HU-3-723, hookline gear, total length of 10.16 m, engine power of 36.77 kw, and gross register tonnage of 6.48 GT), *La Goleta Uno* (plate ID# 3CA-5-6-98, gillnet gear, total length of 9.90 m, engine power of 25.74 kw, and gross register tonnage of 6.59 GT), *Nuevo Río Salado* (plate ID# 3CA-5-10-98, gillnet gear, total length of 9.90 m, engine power of 36.77 kw, and gross register tonnage of 6.59 GT), *Nuevo Ángel Diego* (plate ID# 3CA-5-13-03, gillnet gear, total length of 8.10 m, engine power of 22.06 kw, and gross register tonnage of 2.77 GT) and *Nuevo Hermanos Ariza* (plate ID# 3CA-5-5-00, gillnet gear, total length of 8.05 m, engine power of 16.18 kw, and gross register tonnage of 2.74 GT). All vessels had a capacity for a maximum of 3–4 members. Aboard them, tanks of 30 L with a flow-through system of seawater collected at 2 m depth were installed. The fishing gears were thrown into the water before dawn, from 3.00 to 10.30 h in the morning, and were collected after 2–5 h for the hookline, and after 1.5–4.5 h for the gillnet.

### 2.3. Survival Rates

Survival rates were evaluated according to previous studies conducted in the area with other fishing gears and species [20,23,25,26]. In total, 17 and 18 sets were valid for hookline and gillnet, respectively, and 95 rubberlip grunts were captured (13 to 44 cm furcal length). The minimum number of sets established for this study was 15 per fishing gear, with ≤5 fish per set, to guarantee the data independence that would allow for adequate statistical robustness. Immediately after the arrival of captures to the fishing deck, a maximum of 5 randomly selected individuals per fishing set were introduced in the recovery tanks onboard. They were kept in these tanks until they arrived at the port of Conil, less than 4 h after capture, where they were transferred to independent floating cages (one cage per fishing set and vessel) located at the port entrance. The floating cages had a diameter of 1 m and a depth of 0.60 m, they were protected by isothermal covers (which insulated them from the air temperature), and their walls were made up of a net with a 5 mm mesh size, allowing the water to flow inside the cages. The tidal currents in the area were intense enough to renew the water inside the cages every 6 h. Fish were maintained for a maximum of 48 h in the cages, since it has been described in other fish species that survival after capture can be evaluated after a 24 h recovery [20,23]. Survival was evaluated at 24 and 48 h post-capture by observing the animals: those with normal swimming behavior and breathing rates and lacking serious skin injuries were considered to be alive. Survival rates (as percentages) were calculated for every fishing set, which constitutes the statistical sample for the study. The animals were fasting during these proceedings.

### 2.4. Physiological Recovery

The physiological effects of capture by hookline and gillnet and the recovery responses of rubberlip grunt were evaluated. The 48 h post-capture surviving fish were introduced into chilled seawater at 0 °C (with ice, as this is the method used by local fishermen to slaughter their catch), and they were sampled within less than 3 min from the moment fish perceived their cages being opened. Sampling consisted of measuring the furcal length and body weight, collecting dermal mucus (as a described non-invasive methodology [28,45]), and collecting blood with non-heparinized syringes from the caudal peduncle. Dermal mucus was collected by scraping each animal three times on each side of the body, avoiding the gill area, the anus, and any type of wound that might exist on the skin. All the mucus collected from individual animals was put together in a single Eppendorf vial. Dermal mucus was immediately frozen at −20 °C, while blood was kept on ice. Serum was obtained after centrifuging blood (within less than 2 h after sampling) at 10,000× *g* for 3 min at 4 °C, and frozen at −20 °C. Some animals (*n* = 8–9) were also sampled onboard the fishing vessels to establish the time of maximum stress. As no mortality occurred after 24 h inside the floating cages, some fish were also sampled at this time point to observe possible physiological changes over time once the animals were recovered (*n* = 12 for hookline, and *n* = 21 for gillnet). Once sampled, the fish were returned to the fishermen for human consumption.

### 2.5. Serum and Mucus Measurements

Serum pH was measured immediately after centrifugation with a mini-electrode (HI1083B, Hanna Instruments, Woonsocket, RI, USA). Serum cortisol was measured using an ELISA commercial kit (CAYMAN chemical, ref. No. 500360, Ann Arbor, MI, USA). Serum glucose, lactate, sodium, chloride, calcium, total proteins, phosphate, magnesium and potassium, and dermal mucus glucose, lactate and total proteins were analyzed with commercial kits from Spinreact (Glucose-HK ref. 1001201; Lactate ref. 1001330; Sodium ref. 1001387; Chloride ref. 1001360; Potassium ref. 1001397; Calcium ref. 1001061; Magnesium ref. 1001285; Phosphate ref. 1001155; Total proteins ref. 1001291, Spinreact SA, Sant Esteve de Bas, Spain) as described [35]. Serum and mucus lysozyme and peroxidase activity was measured in duplicate with a 15 µL sample for each parameter, as described in other species [26,32]. All assays were conducted in 96-well microplates and measured using a SPECTROstar Nano spectrophotometer (BMG Labtech, Ortenberg, Germany) using MARS data analysis software for Microsoft Windows.

### 2.6. Growth Parameters and Historical Data

To evaluate the growth of rubberlip grunt, the relationship between total length and weight was calculated according to previous studies [46], using the equation W = a L^b^, where W is the weight (g), L is the length (cm), a = 0.02 and b = 2.94. Growth over time was evaluated according to previous studies [47] using the von Bertalanffy equation L (t) = L∞ [1 − e ^−K (t −to)^], where L(t) is the length (cm), L∞ is the maximum length (80 cm), k = 0.18 represents the von Bertalanffy growth coefficient (year^−1^), t is the age (years), to = −1.038 years and the asymptotic weight is 7.472 kg.

The historical data of catches and sale prices of this species by the artisanal fishing fleet has been provided by the Conil fish market, which regulates all catches in the area. We focused on the captures of the last 7 years (2016 to 2022). A simple model has been developed to evaluate the optimal catch size of this species in which the maximum economic benefits are obtained, based on the survival rates of the discards according to the present study. This model adopts the survival rates obtained in the present study and calculates how much the same animals would be worth if they were recaptured after 0.5, 1 and 1.5 years. For this, the sales prices during the study period are taken into account, and the difference between the original price of the fish sold (historical series) and the price that the same fish would have reached if it had been recaptured after that time is applied. This exercise was carried out for each year of the historical series, and the average of the benefits of the entire period was calculated.

### 2.7. Statistics

Normality and homogeneity of variances were analyzed using the Shapiro-Wilk’s test and the Levene’s test, respectively. The differences in survival rates among fishing gears have been estimated using a Student’s *t*-test. Differences among groups for the physiological parameters related to recovery were tested using two-way ANOVA with group (hookline and gillnet) and time (0, 24 and 48 h) as the factors of variance. Differences between groups for the optimal size of capture were tested using one-way ANOVA with time of recapture after being discarded (0.0, 0.5, 1.0 and 1.5 years) as the factor of variance (sample size was *n* = 7 years of study). When necessary, data were logarithmically transformed to fulfill the requirements of ANOVA. Tukey´s post hoc test was used to identify significantly different groups. Statistical significance was accepted at *p* < 0.05. All the results are given as mean ± s.e.m. (standard error of the mean).

## 3. Results

### 3.1. Survival Rates

As described before for *P. bogaraveo*, *P. mediterraneus* did not float at the surface of the water when released into the recovery tanks, evidencing their ability to quickly regulate the gas bladder. The non-surviving fish did not live past the first 10 h after the introduction in the recovery floating cages. Survival rates up to 48 h were 93.0 ± 3.3% for the gillnet (*n* = 18 fishing sets), and 100.0 ± 0.0% for the hookline (*n* = 17 fishing sets), being statistically different (*p* = 0.046247, Student´s *t*-test). The size of fish captured by gillnet were 23.9 ± 0.5 cm and 224.4 ± 9.9 g (*n* = 57), which was statistically lower (*p* < 0.05, Student’s *t*-test) than the size of those captured by hookline, at 28.1 ± 1.1 cm and 296.2 ± 15.3 g (*n* = 34).

### 3.2. Physiological Recovery after Capture

Serum cortisol (Figure 2A), as a primary stress response in teleosts [33], displayed the highest concentrations immediately after capture (time 0 h), when fish captured by gillnet reached values of 109 ± 16 ng mL^−1^ and those captured by hookline of 43 ± 8 ng mL^−1^, showing significant differences between gears (*p* < 0.05). Cortisol reached asymptotic and lower levels at times 24 and 48 h, without statistical differences between groups (*p* > 0.05). Serum lactate, a paradigmatic secondary stress biomarker in fish [28], is shown in Figure 2B. Serum lactate behaves similarly to cortisol, with higher levels at time 0 h in the gillnet group than in the hookline group, and without significant differences between groups at times 24 and 48 h.

Serum pH, energy metabolites, ions and innate immune system parameters are shown in Table 1. Serum acidification is observed in the moments following capture (reaching circa pH 7.30 with both fishing gears), but the acid-base balance is recovered after 24 h (pH around 7.60, with no statistical differences between any groups at times 24 or 48 h). Phosphate, a pH-buffering molecule, shows significantly different levels between fishing gears at time 0 h with the highest concentration in the gillnet group, followed by insignificantly variable concentrations (2.8 to 3.6 mM) at times 24 and 48 h. Glucose followed the same trend as cortisol and phosphate, with the highest concentrations at time 0 h in the gillnet group compared to the hookline group at the same sampling time, followed by lower and constant values until the end of the assay. Serum protein concentration is highest at time 0 h in both groups, without significant differences between them, followed by lower and constant concentrations. Plasma ion sodium concentrations are around 115 to 155 mM, with the highest values in the hookline group at time 0 h, decreasing its concentration before reaching the lowest values for both groups at time 48 h. There are no statistical differences in serum chloride concentrations between groups or sampling times. Serum calcium and magnesium follow similar trends, with higher values in fish collected by gillnet at time 0 h compared to hookline-captured fish at the same sapling time, and lower values at times 24 and 48 h, without significant differences between groups or times. Serum potassium shows the highest concentrations at time 0 h in both groups, and constant and lower values from then on. Serum peroxidase activity increased at time 48 h in those animals captured by gillnet compared to previous sampling times, while there was no difference among times when captured by hookline. Serum lysozyme activity did not show major changes in any group of fish.

Dermal mucus glucose, lactate, total proteins, peroxidase and lysozyme activity are shown in Table 2. Glucose exhibits opposing patterns in animals caught with each fishing gear, peaking at time 0 h in those captured with hookline, and at times 24 and 48 h in those captured with gillnet. Mucus lactate shows the lowest concentration at time 0 h in all animals, reaching higher but non-significantly different values between groups at times 24 and 48 h. Mucus proteins are significantly increased at time 0 h in those fish caught with hookline, showing the highest concentrations. Mucus peroxidase activity follows the same trend as mucus glucose concentration, with the highest values at time 0 h for the group captured by hookline, and times 24 and 48 h for those animals captured by gillnet. Mucus lysozyme activity shows no differences among sampling times for those fish captured by gillnet, but those captured by hookline reach higher values at time 0 h than at time 24 h, without significant differences between fishing gears at times 24 and 48 h.

### 3.3. Growth Rates and Selling Prices on the Market

In the fish market of Conil, the sizes of the rubberlip grunt are categorized according to their weight as follows: >3 kg (size 1), 2–3 kg (size 2), 1–2 kg (size 3), 0.4–1 kg (size 4) and <400 g (size 5). Total landings per fishing gear and size from the year 2016 to May 2022 are shown in Table 3. The average number of rubberlip grunt landed per year during this period, depending on the size, was: size 1, 3188 ± 1237 kg and 6139 ± 1793 kg for hookline and gillnet, respectively; size 2, 1831 ± 402 kg and 5003 ± 1584 kg for hookline and gillnet; size 3, 7659 ± 2301 kg and 12,746 ± 1948 kg for hookline and gillnet; size 4, 4097 ± 1017 kg and 24,436 ± 6124 kg for hookline and gillnet; and size 5, 1122 ± 349 kg and 11,411 ± 3221 kg for hookline and gillnet. Gillnet captured statistically more fish of sizes 4 and 5 than hookline (*p* < 0.05, Student’s *t*-test).

Considering the growth rate of the species, the sales prices from Table 3, and the economic performance of each fish individual according to their age, Figure 3 shows the relationship between these parameters. It is observed that the maximum price obtained per fish is historically reached with individuals of about 3.5 to 4.5 kg of body weight (size 1).

### 3.4. Economic Benefits in the Case of Discards

As a theoretical exercise, we calculated the possible economic benefits that would be achieved after capture–release–recapture. The considered survival rates for the rubberlip grunts captured by the artisanal fleet of Conil were 100% (according to the results from this study). The calculations are based on what would happen if caught individuals under 3 kg in weight were released (starting from the size they reach the maximum prices). The possible benefits calculated take the following into account: (i) the money obtained from the sale of individuals of each size in the year in which they were captured (original earnings); (ii) the sale price per kg in the year in which they would be recaptured (calculated profits); and (iii) the averaged prices of the entire historical series contemplated in this study. Thus, Figure 4 shows that the maximum economic benefit would occur if all size 4 and 5 individuals (<1 kg in weight) were released back into the ocean and recaptured for sale after 1 to 1.5 years (reaching 1–1.8 kg per fish). Adding the sales of these recaptured animals that would otherwise be sold as sizes 4 and 5 would give around 234 to 960% of benefits per year, which would result in an additional EUR 87–323 thousand per year. Moreover, those animals of size 3 that were around 1 kg of weight would increase 15% in body mass after 6 months in the wild, and this would then result in economic benefits of 26% in the market (around EUR 30,000 extra per year).

## 4. Discussion

The present study takes advantage of the physiological tools developed to ensure survival and recovery of discards and applies them to increase the economic benefits of artisanal fisheries. Thus, rubberlip grunt captured by hookline and gillnet in southern Europe shows survival rates above 93%, which may facilitate their exemption to the European landing obligation. Contemplating the possibility of recapturing discards, we described that maximum economic benefits occur if only animals above 1 kg of body weight are captured.

### 4.1. Survival of Discards

*P. mediterraneus* shows no buoyancy problems associated with decompression when captured between 10 to 50 m depth. Hence, animals captured with hookline and gillnet by the fleet of Conil (South of Spain) can swim and dive immediately afterwards. Similar results were observed in *P. bogaraveus* captured in the same geographical area at depths around 234 to 452 m by hookline [20]. This ability to regain buoyancy is crucial to improving the survival of captured species, as barotrauma greatly diminishes survival rates of captured teleosts [48,49]. In this sense, the present study shows survival rates higher than 93% when captured by gillnet, and up to 100% when *P. mediterraneus* is captured by hookline. These differences may be associated with the possibility that fish caught by gillnet suffer skin injuries due to rubbing against the net, as well as death by suffocation due to the operculum getting caught in the net. However, both fishing gears show higher survival rates than other species captured by different fishing gears. In this sense, *P. bogaraveo* captured by the fishing hookline gear called *voracera* in the Strait of Gibraltar, which represents the first exemption to the European landing obligation [20], showed survival rates of 90.6%, which is lower than the results from the present study. Flatfish species captured by bottom trawling in northern Europe, such as *Pleuronectes platessa*, *Solea solea* and *Limanda limanda* reach survival rates between 2 and 58% [22,50], which are much lower than those described in the present study. European pilchard (*Sardina pilchardus*) captured by purse seine in southern Portugal show estimated survival rates of up to 44.7%, showing that scale loss can be a good indicator of the damage suffered by the fish [21]. Moreover, herring (*Clupea harengus*), also captured by purse seine in the North Sea, had a survival rate of up to 50%, which depends on the crowding density and the physical damage suffered by the animals in the fishing process [36]. Fishing gear is thus a relevant factor affecting the survival of fish, although the catshark *Scyliorhinus canicula* captured by bottom-trawling in the same waters as those of the present study shows survival rates up to 95.7% [23]. An important aspect to take into account in order to approve the exemptions to the landing obligation is to know the number of animals that would be affected by this fact. For this reason, in the present study it was calculated that around 2000 fish of size 5 (mean calculated during the entire study period), and less than 3000 animals of size 4 would be affected by the release. In addition to the fishing gear used, the amount of time that the animals are exposed to the air is an important factor that affects survival, as has been described in chum salmon (*Oncorhynchus keta*) captured by purse seine on the coast of British Columbia, in Canada [51], or *S. canicula* captured by bottom-trawling in the Gulf of Cadiz, in the waters of southeastern Europe [23]. In the present study, exposure to air is very brief (less than one minute) since the animals hoisted on board are immediately separated by hand from the fishing gear, which may favor the survival. At-vessel immediate mortality has also been evaluated in other elasmobranchs such as rays, captured by longline, otter trawl, drift trammel net and tangle net in the North Sea and the English Channel, and ranged from 0 to 6.4% [24], although further studies are needed to assess the medium-term survival of these species. In this sense, the approach defined in the present study and described before for other fish species and aquatic animals [20,21,23,25,26,52,53], in which survival associated with physiological recovery is evaluated, is useful to reduce the observation times of the animals in captivity and to ensure their complete recovery.

### 4.2. Physiological Recovery after Capture

The capture of wild animals is a dramatic event that represents an episode of acute stress, as confirmed by the high levels of serum cortisol in *P. mediterraneus* captured in this study. Thus, fishing induces a series of physiological responses that can serve as markers to determine the degree of distress on the fish [40]. Alongside primary stress responses (such as cortisol), secondary responses are observed in the present study, highlighting differences in the degree of affliction on the fish captured by each of the fishing gears, which can explain the differences in their survival rates. In this regard, after an acute-stress challenge, cortisol levels reach maximum concentrations in the blood after 30 to 60 min after the stressful process begins, decreasing to basal values within less than 6 h [20,28,38,54]. The maximum concentration of cortisol also depends on the nature of the stressor itself and the environmental conditions [28,55,56].

In our study, the highest cortisol levels are three times the baseline values, aligning with what was observed in *P. bogaraveo* captured by hookline [20], *Sparus aurata* and *Colossoma macropomum* exposed to air [28,38], or *Acanthopagrus australis* captured by trawling [56]. Other fish species reach higher cortisol levels compared to unstressed animals. Hence, *S. pilchardus* captured by purse seine show five-fold more cortisol after capture [21], cortisol concentrations increased eight-fold after scale removal in *C. harengus* [57], or even extreme responses can be observed, such as the dramatic 30-fold increase in plasma cortisol after air exposure observed in *Solea senegalensis* [54]. In other fish taxa, such as catsharks, air exposure induced a six-fold increase in plasma 1α-hidroxicorticosterone [32,58], which is the glucocorticoid hormone in elasmobranches [32]. Altogether, acute stress responses are species-specific and should be studied separately if robust results that can be used for fisheries management are to be obtained. In the present study, higher serum cortisol levels observed in *P. mediterraneus* captured by gillnet compared to those captured by hookline may indicate: (i) that gillnet represents a more aggressive capture method for this species, which may also be related to the lower survival rates; and/or (ii) that fish captured by hookline have been attached longer to the hook than those caught by net, indicating that in Figure 2 we observe the curve of decline in cortisol levels. The analysis of the secondary responses to stress is, therefore, necessary to be able to know the degree of affliction of the fish.

In our study, secondary stress responses such as energy metabolites, acid–base balance, and osmoregulatory imbalances in serum occur in all fish immediately after capture (time 0 h). However, all measured stress biomarkers reached an asymptote from 24 h onwards, indicating a physiological recovery of the survivors, as seen before in other fish species [20,23], as well as in crustaceans [26] or even cephalopods [25]. It should be mentioned that *P. mediterraneus* captured by gillnet show increased serum levels of glucose, lactate, phosphate, calcium and magnesium, and decreased values of sodium, immediately after capture, coinciding with higher serum cortisol concentrations. These results support that gillnet is more stressful for this species than hookline, and hence induced lower survival rates.

Reinforcing this idea, this study found that the immune system response in dermal mucus, especially peroxidase activity, increases over time in gill-caught fish. These results may be related to skin lesions caused by gill nets, which activated the immune response, although no major wounds were observed in these animals. *S. aurata* and *Dicentrarchus labrax* increased their peroxidase activity 24 h after infection by *Photobacterium damselae* [59,60]. Supporting this information, mucus glucose, as a good parameter reflecting the skin mucus response in teleosts [45], also exhibits higher concentration in *P. mediterraneus* caught by gillnet at the end of the experimental period. However, the changes observed in the dermal mucus are minor considering the general physiology of the fish studied, so it can be stated with confidence that the surviving fish in the present study recover their physiological homeostasis within the first 24 h after capture if they are maintained in proper conditions.

### 4.3. Economic Benefits Due to the Recapture of Discards

Current European legislation establishes that captured fish regulated by minimum sizes or quotas, such as the rubberlip grunt, must be landed (European Regulation N° 1380/2013). The minimum size for its marketing in Spain is 20 cm (140 g body weight). According to our results, those individuals below 1 kg weight represent little benefit for the fisherman compared to larger animals (less than EUR 3.56 kg^−1^ compared to more than EUR 5.68 kg^−1^). *P. mediterraneus* requires 2.8 years to reach 1 kg, and from that weight on, the growth curve of the species becomes exponential. Thus, it seems reasonable to propose the discarding of animals below that size to recapture them a few months later, when they will have reached higher market values per kg. Further tagging and release studies are required to explore the possible migrations of this species, as well as the actual survival in the natural environment. For this theoretical proposal, it would be necessary to know the natural mortality of the species in order to have a more accurate idea about the economic evolution of the fishery. This study is based on actual sales data from recent years, as well as very conservative estimates of the growth of the species, and realistic estimates based on its survival if discarded. This would yield profit margins that could improve the economy of the fishing fleet in the geographical area of study (Conil, Spain), but would require a holistic commitment on the part of all stakeholders and people involved in this fishery.

### 4.4. Future Strategies to Improve Artisanal Fisheries

Improving small-scale fisheries requires addressing data gaps by engaging the fisheries sector with science through innovative approaches [61]. To the best of our knowledge, this is the first approach that virtually links survival of discards to improvements in the fish market when recaptured. Other studies have focused on the effects of fishing on phenotypic selection of wild individuals [62]. Artisanal fisheries must make changes in their management strategies in order to be profitable in the long term, as they are experiencing a deep crisis due to reduced catches [15]. It is therefore evident that small-scale fisheries must be differentiated from other seafood producers such as large-scale fisheries or aquaculture [63]. During recent years, numerous actions have been carried out in Europe to improve and differentiate artisanal fishery products, such as the creation of quality seals and certificates (associated to marketing strategies), or the direct sale of fishery products to the final consumer (short supply chains) [63]. The creation of fishing groups, where a series of (self-imposed) rules are followed for the common improvement of fishermen´s conditions, is useful for the progress of the sector. In this sense, the Organization of Fishery Producers OPP72 of Conil is a paradigmatic example of success not only in Spain, but in the whole of Europe. This organization currently leads international small-scale fishing groups aiming for the improvement of the artisanal fishing sector. In the case of obtaining the exemption from the landing obligation for *P. mediterraneus* (or any other species whose survival and recovery is scientifically proven), the OPP72 would be the right organization to manage its fishery following the economic criteria of maximization of benefits based on scientific studies similar to this one.

Thus, the present study describes the survival rates of *P. mediterraneus*, a highly appreciated fish species captured in Europe by artisanal fisheries, with the economic benefits that their recapture would entail once they have reached a more profitable size. It is shown that gillnet and hookline fishing gears, under the methodologies described here, achieve survival rates of rubberlip grunt of almost 100%. Future studies should be directed to other species of economic interest, to achieve their exemption from the European landing obligation, as well as reaching agreements between fishermen to respect the size limits imposed according to the scientific criteria described here.

## 5. Conclusions

This study collaborates in the development of artisanal fishing in southern Europe. The high survival of *P. mediterraneus* is hereby demonstrated, and the information will be made known to the management bodies.

## Figures and Tables

**Figure 1 animals-12-03423-f001:**
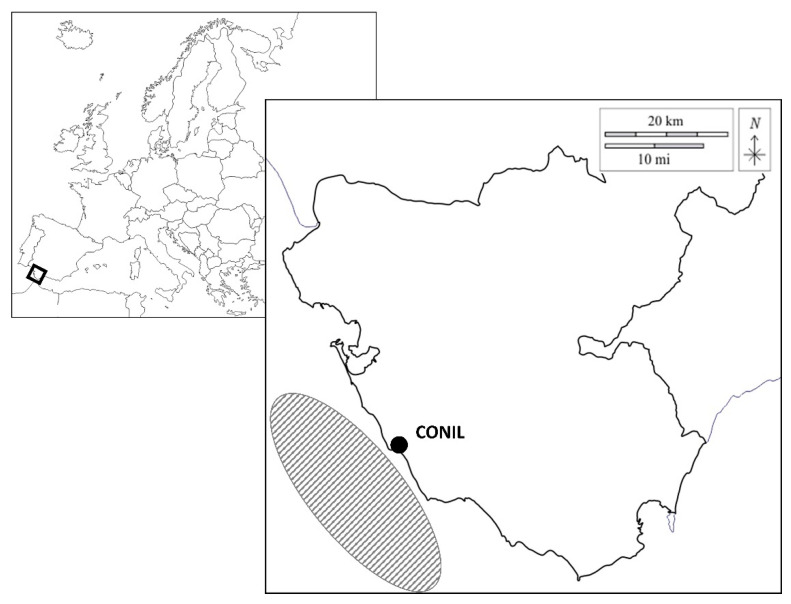
Map of Europe (top left) where the province of Cadiz (Spain) is marked with a black rectangle. Enlargement of the map of the Cadiz province (bottom right) highlighting the port of Conil with a dot. Artisanal fishing area of the Conil fleet (gray striped oval).

**Figure 2 animals-12-03423-f002:**
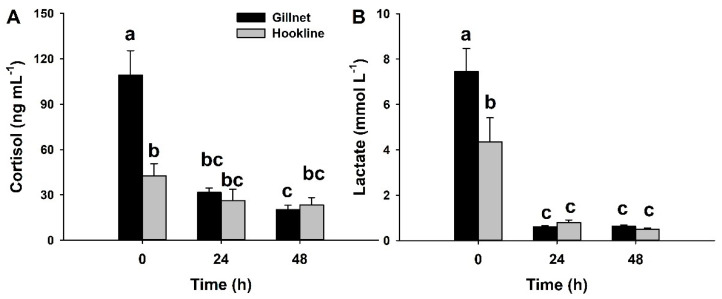
Serum cortisol (**A**) and lactate (**B**) in *P. mediterraneus* captured by gillnet (black bars) or hookline (grey bars). Fish were sampled immediately after capture (0 h) or at times 24 h and 48 h after their introduction into floating sea cages. Data are expressed as mean + SEM (*n* = 8–25 per column). Different lowercase letters indicate significantly different groups (*p* < 0.05, two-way ANOVA followed by a post hoc Tukey test).

**Figure 3 animals-12-03423-f003:**
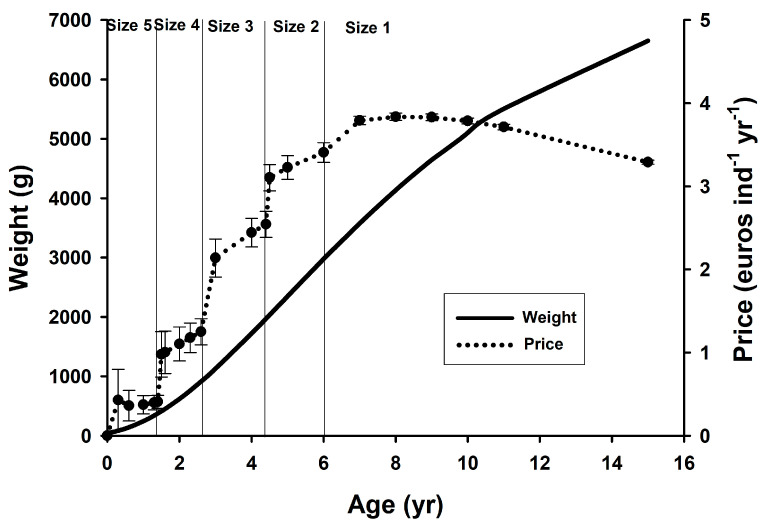
Growth of *P. mediterraneus* according to the published literature [46,47] (solid line), and selling prices per individual, i.e., the price that was obtained when it was sold divided by their age calculated through the von Bertalanffy equation (dotted line, mean ± SEM representing the differences between the years used for the calculation). Vertical lines represent the limits of the commercial sizes (1 to 5) in the fish market of Conil (south of Spain).

**Figure 4 animals-12-03423-f004:**
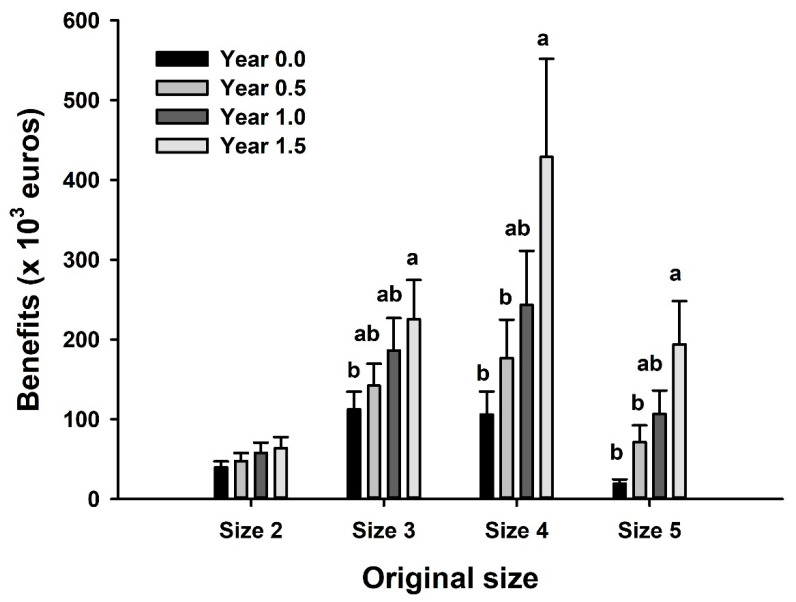
Averaged sale price in the fish market of rubberlip grunt caught in Conil since 2016 (until May 2022) depending on the commercial size (year 0.0). The figure shows the price that these same individuals would have reached if they had been released and further recaptured after 0.5, 1.0 and 1.5 years. Columns show mean + SEM (*n* = 7 years). Different letters indicate significant differences among groups of the same commercial size (*p* < 0.05, one-way ANOVA followed by a Tukey post hoc test).

**Table 1 animals-12-03423-t001:** Serum pH, energy metabolites, and ions in *P. mediterraneus* captured by gillnet or hookline. Fish were sampled immediately after capture (0 h) or at times 24 h and 48 h after their introduction into floating sea cages. Data are expressed as mean ± SEM (*n* = 8–25 per group and time). Different lowercase letters indicate significantly different groups (*p* < 0.05, two-way ANOVA followed by a post hoc Tukey test).

Parameter	Fishing Gear	0 h	24 h	48 h
pH	Gillnet	7.26 ± 0.03 b	7.63 ± 0.01 a	7.59 ± 0.01 a
	Hookline	7.35 ± 0.06 b	7.55 ± 0.02 a	7.60 ± 0.02 a
Phosphate (mM)	Gillnet	7.2 ± 0.5 a	3.6 ± 0.2 bc	2.8 ± 0.2 c
	Hookline	4.8 ± 0.6 b	3.2 ± 0.2 c	2.8 ± 0.1 c
Glucose (mM)	Gillnet	7.9 ± 0.9 a	3.4 ± 0.4 bc	2.9 ± 0.2 c
	Hookline	5.0 ± 0.5 b	3.9 ± 0.4 bc	2.9 ± 0.1 c
Total proteins (g dL^−1^)	Gillnet	4.2 ± 0.4 a	2.4 ± 0.1 b	2.7 ± 0.1 b
	Hookline	3.6 ± 0.2 a	2.5 ± 0.1 b	2.8 ± 0.1 b
Na^+^ (mM)	Gillnet	130 ± 6 ab	140 ± 9 a	114 ± 4 b
	Hookline	154 ± 4 a	140 ± 8 a	129 ± 3 ab
Cl^−^ (mM)	Gillnet	182 ± 5	194 ± 2	191 ± 3
	Hookline	189 ± 4	190 ± 2	188 ± 1
Ca^2+^ (mM)	Gillnet	3.6 ± 0.1 a	2.3 ± 0.1 b	2.2 ± 0.0 bc
	Hookline	2.5 ± 0.1 b	2.2 ± 0.1 bc	2.0 ± 0.1 c
Mg^2+^ (mM)	Gillnet	3.5 ± 0.4 a	2.1 ± 0.2 b	2.0 ± 0.2 b
	Hookline	1.7 ± 0.5 b	1.1 ± 0.2 b	1.3 ± 0.2 b
K^+^ (mM)	Gillnet	7.4 ± 0.4 a	4.7 ± 0.3 b	4.1 ± 0.3 bc
	Hookline	6.6 ± 0.9 a	3.6 ± 0.5 c	4.1 ± 0.3 bc
Peroxidase activity (U mL^−1^)	Gillnet	97 ± 16 ab	68 ± 12 b	109 ± 9 a
	Hookline	107 ± 18 ab	88 ± 13 ab	91 ± 9 ab
Lysozyme activity (µg mL^−1^)	Gillnet	8.9 ± 0.9 ab	7.0 ± 0.7 b	10.7 ± 1.1 ab
	Hookline	9.7 ± 1.0 ab	9.3 ± 1.6 ab	12.7 ± 1.5 a

**Table 2 animals-12-03423-t002:** Dermal mucus glucose, lactate and total proteins in *P. mediterraneus* captured by gillnet or hookline. Fish were sampled immediately after capture (0 h) or at times 24 h and 48 h after their introduction into floating sea cages. Data are expressed as mean ± SEM (*n* = 8–25 per group and time). Different lowercase letters indicate significantly different groups (*p* < 0.05, two-way ANOVA followed by a post hoc Tukey test).

Parameter	Fishing Gear	0 h	24 h	48 h
Glucose (mM)	Gillnet	0.46 ± 0.06 b	0.63 ± 0.06 a	0.65 ± 0.04 a
	Hookline	0.72 ± 0.06 a	0.37 ± 0.04 b	0.45 ± 0.04 b
Lactate (mM)	Gillnet	0.14 ± 0.02 c	0.62 ± 0.07 a	0.45 ± 0.04 ab
	Hookline	0.15 ± 0.04 c	0.43 ± 0.08 abc	0.30 ± 0.03 bc
Total proteins (g dL^−1^)	Gillnet	0.86 ± 0.17 b	0.66 ± 0.08 b	0.91 ± 0.07 b
	Hookline	1.60 ± 0.36 a	0.45 ± 0.10 b	0.77 ± 0.07 b
Peroxidase activity (U mL^−1^)	Gillnet	56 ± 10 bc	87 ± 7 ab	104 ± 8 a
	Hookline	87 ± 12 ab	57 ± 13 bc	38 ± 4 c
Lysozyme activity (µg mL^−1^)	Gillnet	0.47 ± 0.15 ab	0.66 ± 0.11 ab	0.56 ± 0.11 ab
	Hookline	1.17 ± 0.41 a	0.18 ± 0.09 b	0.55 ± 0.09 ab

**Table 3 animals-12-03423-t003:** Total landings and selling prices of *P. mediterraneus* captured by hookline and gillnet by the artisanal fleet of Conil (south of Spain) from 2016 to May 2022. It is observed that the price increases with fish size, and that price varies over the years.

Year	Size	Total Captures (kg)		Total Sales	
		Hookline	Gillnet	(Euros)	(Euros kg^−1^)
2016	1	8605	11,767	128,263	6.29
	2	3739	10,024	67,235	4.89
	3	6344	17,500	87,448	3.67
	4	4544	10,254	33,639	2.27
	5	1101	2975	4723	1.16
2017	1	7199	13,036	129,490	6.40
	2	2280	10,208	54,757	4.38
	3	4077	12,136	55,819	3.44
	4	2051	9503	23,968	2.07
	5	328	4742	6187	1.22
2018	1	1118	7726	63,405	7.14
	2	928	7873	54,120	6.15
	3	3246	17,341	107,481	5.21
	4	2692	33,610	147,958	4.06
	5	760	20,405	33,509	1.57
2019	1	1435	3918	37,465	6.96
	2	1363	2534	27,570	7.03
	3	5705	14,915	119,640	5.80
	4	3352	46,384	181,418	3.64
	5	955	23,478	38,577	1.58
2020	1	1289	1578	24,369	8.50
	2	2034	1915	29,997	7.60
	3	11,141	14,116	151,761	6.01
	4	5284	30,454	124,285	3.47
	5	1363	12,537	21,663	1.56
2021	1	2033	3071	41,696	8.17
	2	1997	1867	32,410	8.39
	3	19,978	10,690	216,156	7.05
	4	9365	36,494	201,368	4.39
	5	3024	13,670	25,371	1.52
2022 (to May 28)	1	640	1874	21,234	8.44
	2	474	601	10,361	9.64
	3	3121	2524	48,635	8.61
	4	1390	4353	28,913	5.03
	5	324	2069	4911	2.05

## Data Availability

Not applicable.

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
