# Peer review of "Economic Improvement of Artisanal Fishing by Studying the Survival of Discarded Plectorhinchus mediterraneus"

_animals, 2022, doi:10.3390/ani12233423_

Round 1
Reviewer 1 Report
Dear Authors
The article submitted for review is interesting. I recommend minor revisions to the manuscript before possible publication:
1. Please complete the Simple Summary section; as it stands, it is unsatisfactorily written.
2. Please follow the guidelines for Authors on how to cite articles in Animals journal in References section.
Regards
Author Response
Dear Authors
The article submitted for review is interesting. I recommend minor revisions to the manuscript before possible publication:
- Please complete the Simple Summary section; as it stands, it is unsatisfactorily written.
Authors: Done, as suggested.
- Please follow the guidelines for Authors on how to cite articles in Animals journal in References section.
Authors: We have revised the references to adapt them to the journal. Thanks for the appreciation.
Regards
Reviewer 2 Report
Unfortunately, for ethical – i.e., sustainability and animal welfare reasons – I must recommend the rejection of this manuscript. The European Regulation EU N° 1380/2013 aims at the preservation of marine biological resources and the management of fisheries targeting them. However, this paper supports a strategy that would not only harm many individual fish, but that would impair the essential goals of the regulation. Even if the reported survival rate would prove correct, the mortality percentage estimated by the researchers would imply that a very large number of animals would die per year as a consequence of the proposed exemption, and thus put at risk this the goals of this EU regulation whose goal is to make progress with preserving key marine biological resources. Instead of lowering the bar for this type of fisheries, I would recommend the investigation of alternative strategies that might help this industry to effectively comply with the regulation.
Author Response
Comments and Suggestions for Authors
Unfortunately, for ethical – i.e., sustainability and animal welfare reasons – I must recommend the rejection of this manuscript. The European Regulation EU N° 1380/2013 aims at the preservation of marine biological resources and the management of fisheries targeting them. However, this paper supports a strategy that would not only harm many individual fish, but that would impair the essential goals of the regulation. Even if the reported survival rate would prove correct, the mortality percentage estimated by the researchers would imply that a very large number of animals would die per year as a consequence of the proposed exemption, and thus put at risk this the goals of this EU regulation whose goal is to make progress with preserving key marine biological resources. Instead of lowering the bar for this type of fisheries, I would recommend the investigation of alternative strategies that might help this industry to effectively comply with the regulation.
Authors: We are very sorry to disagree with the reviewer. We agree with the vision of preserving the ecosystem. However, this scientific study should be refuted with scientific arguments to exclude it from publication. The authors emphasize that, if Europe were to approve the release of P. mediterraneus (which depends on many factors beyond our control), it would be limited to this species, in the geographical location detailed here, and with the artisanal fishing gears that are described in the work. The number of animals per year that could be released (in the hypothetical case that the exemption is approved), and die, would not be as high as the reviewer suggests. Moreover, it would be a much lower number of animals than other de minimis exemptions from other fisheries. However, it could help preserve the artisanal fishery in the area, which has been proven to be much more respectful to the environment and the ecosystem than other fisheries. With all this, we deeply respect the reviewer's decision, although we do not agree with it.
Reviewer 3 Report
It is an interesting study with potential impact. I made my comments on the manuscript.
Most comments are related to style and choice of words, and I suggested some alternatives for clarity. I also made some suggestions about the language, and the authors must please check whether these changed the intended meaning.
I am not able to understand the label of the Fig on page 12. I suspect that an expanded explanation in the label will support understanding and I made some comments in the sections
I am not clear on whether the same fish specimen was scraped more than once for mucus collection. If that has happened, it may explain the increased levels. Please clarify in methods and results.
Author Response
Comments and Suggestions for Authors
It is an interesting study with potential impact. I made my comments on the manuscript.
Authors: We could not find in the webpage the file with the modifications.
Most comments are related to style and choice of words, and I suggested some alternatives for clarity. I also made some suggestions about the language, and the authors must please check whether these changed the intended meaning.
Authors: The entire manuscript has been reviewed by a native English speaker.
I am not able to understand the label of the Fig on page 12. I suspect that an expanded explanation in the label will support understanding and I made some comments in the sections
Authors: The label has been modified for clarity, and some information has been added in the M&Ms section.
I am not clear on whether the same fish specimen was scraped more than once for mucus collection. If that has happened, it may explain the increased levels. Please clarify in methods and results.
Authors: All animals were sampled one time each. We have included more information about dermal mucus collection in the M&Ms section.
Reviewer 4 Report
This paper focuses on the landing obligation (LO) recently imposed by the EU to end wasteful discards of fish by EU vessels. The LO has a negative effect of small-scale fishers’ profits because it forces them to land fish which they are not targeting or have no quota for or have little commercial value. In effect, such fish are being discarded on land instead of at sea. To avoid such regrettable consequences, the EU allows some exceptions to the discard ban, including discard survivability. If scientific evidence could prove that particular discarded species could survive being discarded at sea, fishers would be allowed to discard them. This study examines the survival rates of discarded rubberlip grunt fish in Spain, linked to its market prices, and concludes that allowing discards of these fish less than one kg in weight, would benefit both the future stock and the livelihoods of fishers.
My view is that this is an excellent paper, deserving publication as it stands. I am a social scientist, however, not a natural scientist, so I cannot vouchsafe for the scientific validity of the methods used for measuring the survivability rates of the rubberlip grunt fish in the experiments you carried out. Nevertheless, from my experience as a social scientist with marine scientists carrying out similar discard survivability experiments in UK waters, the methods employed here seem very rigorous.
Author Response
Comments and Suggestions for Authors
This paper focuses on the landing obligation (LO) recently imposed by the EU to end wasteful discards of fish by EU vessels. The LO has a negative effect of small-scale fishers’ profits because it forces them to land fish which they are not targeting or have no quota for or have little commercial value. In effect, such fish are being discarded on land instead of at sea. To avoid such regrettable consequences, the EU allows some exceptions to the discard ban, including discard survivability. If scientific evidence could prove that particular discarded species could survive being discarded at sea, fishers would be allowed to discard them. This study examines the survival rates of discarded rubberlip grunt fish in Spain, linked to its market prices, and concludes that allowing discards of these fish less than one kg in weight, would benefit both the future stock and the livelihoods of fishers.
My view is that this is an excellent paper, deserving publication as it stands. I am a social scientist, however, not a natural scientist, so I cannot vouchsafe for the scientific validity of the methods used for measuring the survivability rates of the rubberlip grunt fish in the experiments you carried out. Nevertheless, from my experience as a social scientist with marine scientists carrying out similar discard survivability experiments in UK waters, the methods employed here seem very rigorous.
Authors: We deeply appreciate the words of the reviewer.
Reviewer 5 Report
The conclusion is discussing and not finalizing the work.
Author Response
Comments and Suggestions for Authors
The conclusion is discussing and not finalizing the work.
Authors: The Conclusion section has been modified accordingly.
Round 2
Reviewer 2 Report
Regarding my previous considerations on the manuscript, it is relevant to emphasize that they are ultimately based on scientific aspects related to the implications of the fishing practices examined by the researchers in terms of animal welfare and sustainability. As I mentioned, even if the short-term survival rate provided by the authors were a realistic estimate, those individuals who die as a result of these practices would certainly endure poor welfare leading to their deaths. On top of that, if the goal of the study is to evaluate the post-catch survival of these animals, the study should also investigate the longer term survival (post-release) to fully understand the actual implications in terms of welfare and survival of the physiological aggressions that these animals experience as part of the practices that they are subjected to. The authors briefly mention by the end of the manuscript (lines 456-457) that “further tagging and release studies are required to explore the possible migrations of this species, as well as the actual survival in the natural environment.” This is a very important point and should be a central piece of the study when the goal is to evaluate the actual post-cath survival of these individuals in relation to a possible exemption of the European Regulation EU N° 1380/2013.
Finally, the extent to which these practices might challenge the sustainability of the local wild population of this species at least warrant further discussion in the manuscript. Despite mentioning potential benefits of artisanal fisheries regarding the sustainability of the species, the authors are not even providing estimates for the numbers of animals that would be negatively impacted by this type of fisheries – provided there was an exemption of the EU regulation for this species in the investigated geographical area.
For all these reasons I must again recommend the rejection of this manuscript. Considering the serious consequences of the potential exemption mentioned by the authors, I encourage them to pay attention to the additional aspects mentioned above in order to provide more compelling evidence regarding the question they are tackling.
Author Response
Comments and Suggestions for Authors
Regarding my previous considerations on the manuscript, it is relevant to emphasize that they are ultimately based on scientific aspects related to the implications of the fishing practices examined by the researchers in terms of animal welfare and sustainability. As I mentioned, even if the short-term survival rate provided by the authors were a realistic estimate, those individuals who die as a result of these practices would certainly endure poor welfare leading to their deaths. On top of that, if the goal of the study is to evaluate the post-catch survival of these animals, the study should also investigate the longer term survival (post-release) to fully understand the actual implications in terms of welfare and survival of the physiological aggressions that these animals experience as part of the practices that they are subjected to. The authors briefly mention by the end of the manuscript (lines 456-457) that “further tagging and release studies are required to explore the possible migrations of this species, as well as the actual survival in the natural environment.” This is a very important point and should be a central piece of the study when the goal is to evaluate the actual post-cath survival of these individuals in relation to a possible exemption of the European Regulation EU N° 1380/2013.
Authors: We fully agree with the Reviewer that post-release studies must be conducted. That is why we are actually carrying out a tagging program for the animals captured under the conditions described in this manuscript, although this information is not reflected on it.
Finally, the extent to which these practices might challenge the sustainability of the local wild population of this species at least warrant further discussion in the manuscript. Despite mentioning potential benefits of artisanal fisheries regarding the sustainability of the species, the authors are not even providing estimates for the numbers of animals that would be negatively impacted by this type of fisheries – provided there was an exemption of the EU regulation for this species in the investigated geographical area.
Authors: We appreciate the comment. The results of the survival study are actually providing an estimate of the number of animals that will be negatively affected by the (hypothetical future) release. We have now included this information in the Discussion section.
For all these reasons I must again recommend the rejection of this manuscript. Considering the serious consequences of the potential exemption mentioned by the authors, I encourage them to pay attention to the additional aspects mentioned above in order to provide more compelling evidence regarding the question they are tackling.
Authors: We appreciate the honesty of the Reviewer.
Reviewer 3 Report
I made a few minor suggestions to improve the readability. I am satisfied that the authors attended to the required amendments

Author Response
Comments and Suggestions for Authors
I made a few minor suggestions to improve the readability. I am satisfied that the authors attended to the required amendments.
Authors: We deeply appreciate the Reviewer´s efforts and all suggestions have been included in the manuscript.